# Expression of Long Noncoding RNAs in Fibroblasts from Mucopolysaccharidosis Patients

**DOI:** 10.3390/genes14020271

**Published:** 2023-01-20

**Authors:** Zuzanna Cyske, Lidia Gaffke, Karolina Pierzynowska, Grzegorz Węgrzyn

**Affiliations:** Department of Molecular Biology, Faculty of Biology, University of Gdansk, Wita Stwosza 59, 80-308 Gdansk, Poland

**Keywords:** mucopolysaccharidoses, long noncoding RNAs, gene expression regulation

## Abstract

In this report, changes in the levels of various long non-coding RNAs (lncRNAs) were demonstrated for the first time in fibroblasts derived from patients suffering from 11 types/subtypes of mucopolysaccharidosis (MPS). Some kinds of lncRNA (*SNHG5*, *LINC01705*, *LINC00856*, *CYTOR*, *MEG3*, and *GAS5*) were present at especially elevated levels (an over six-fold change relative to the control cells) in several types of MPS. Some potential target genes for these lncRNAs were identified, and correlations between changed levels of specific lncRNAs and modulations in the abundance of mRNA transcripts of these genes (*HNRNPC, FXR1*, *TP53*, *TARDBP*, and *MATR3*) were found. Interestingly, the affected genes code for proteins involved in various regulatory processes, especially gene expression control through interactions with DNA or RNA regions. In conclusion, the results presented in this report suggest that changes in the levels of lncRNAs can considerably influence the pathomechanism of MPS through the dysregulation of the expression of certain genes, especially those involved in the control of the activities of other genes.

## 1. Introduction

Lysosomal storage disease (LSD) constitutes a group of disorders caused by a lack of the activity of specific lysosomal enzymes (such as acid hydrolases), membrane proteins, and deficiencies in lysosomal transporters or signals required for delivery of an enzyme to the lysosome. This leads to a deficiency in the lysosomal degradation of various macromolecules, and, in turn, to their accumulation in lysosomes [1,2]. 

Mucopolysaccharidoses (MPS) are a group of LSDs caused by a significant deficiency in the activity of lysosomal enzymes involved in the degradation of glycosaminoglycans (GAGs) [3]. GAGs are long, non-branched polysaccharides that are degraded due to their cleavage by endohexosidases or exoglucuronidases, followed by the removal of their shorter fragments by the subsequent actions of other enzymes [3]. The actions of all GAG-degrading enzymes are correlated and sequential, i.e., the next enzyme can only act after the action of the previous one. Thus, if only one enzyme is deficient, the whole degradation pathway is highly impaired.

There are 13 types/subtypes of MPS occurring in humans (plus one that, to date, has only been identified in an animal model), which are classified according to a lack or deficiency of a specific enzyme and the kind of stored GAG(s) [4,5]. These diseases are inherited in an autosomal recessive manner, except MPS II, which is a sex-linked disease. MPS is a rare disease, but its specific prevalence depends on the particular type/subtype. For example, MPS I occurs at a frequency of 1 per 88,000 live births, while there are only 4 cases of MPS IX described in the literature [6]. There are some MPS symptoms occurring in all or most types, such as coarse facial features, organomegaly, and changes in the cardio-vascular system and bones. However, there are also symptoms specific for particular types, such as mental deterioration in some clinical forms of MPS I and II and in all patients with MPS III [4,5,7].

Every MPS type is a severe disease, and each significantly shortens life spans (up to 2 decades on average) [6,7,8]. Although some therapeutic options are available, such as enzyme replacement therapy (ERT) or hematopoietic stem cell transplantation, and even though other therapeutic approaches are being investigated [9], the treatment of all the symptoms of patients cannot be achieved despite reductions in GAG levels [10,11]. This suggests that there are other—perhaps secondary and tertiary—causes of cellular dysfunctions in MPS cells. Recent reports have indicated that not only the functions of lysosomes, but also other cellular processes are disturbed in patients with this disease. These include dysfunctions of the cytoskeleton and mitochondria [12], the disturbed regulation of apoptosis, and impaired autophagy [13,14]. However, studies on the secondary and tertiary effects of MPS were not extensive, and our knowledge in this field is limited [15]. Nevertheless, the results of such investigations indicated that there is a need for research into additional pathomechanisms of MPS that may indicate novel therapeutic targets.

One of the secondary pathomechanisms of MPS constitutes changes in the expression of certain genes [16]. The products of such genes can include regulatory proteins, transcription factors, and other molecules involved in various biochemical reactions. Changes in their levels may negatively influence cellular functions and thus cause perturbations at the level of the organism. Such phenomena have been demonstrated recently and include gene expression dysregulation-related disturbances of cell activation [17], changes in proteasome composition and activity [18], alterations in the modulation of apoptosis [19], abnormalities in the morphologies of organelles and cellular components’ organization [20], impaired ion homeostasis [21], disturbances in signal transduction [22], and the dysregulation of the cell cycle [23]. 

Among the various possible causes of the disturbed regulation of gene expression, there are the actions of non-coding RNA (ncRNA) molecules. Such RNAs include small interfering RNAs (siRNAs), piwi-interacting RNAs (piRNAs) [24], micro RNAs (miRNAs) [25], long non-coding RNAs (lncRNAs) [26], circular RNAs (circRNAs) [27], and others. The ncRNA species do not encode proteins but influence gene expression and may affect modifications of DNA and histones [28,29]. 

The group of lncRNAs consists of differentiated transcripts of lengths exceeding 200 nucleotides [26]. Various lncRNAs were identified and classified into sub-classes, depending on their lengths and functions [30]. Some lncRNAs are encoded by DNA fragments located between protein-coding genes; thus, they are called long intergenic non-coding RNAs (lincRNAs) [31]. Other lncRNAs, which are particularly long (over 50 kb), are called very-long intergenic non-coding RNAs [32]. Generally, lncRNAs influence gene expression by two different pathways. First, they bind to miRNAs to sponge them and prevent their interactions with target mRNAs. This causes enhanced synthesis of the proteins encoded by the target genes. Second, lncRNAs can interact directly with target mRNAs, thus modulating their translation. Therefore, lncRNAs can significantly down- or up-regulate the expression of target genes [33,34]. 

Recently, there has been an increasing number reports on the roles of ncRNAs in the pathogenesis of many diseases, including neurodegenerative diseases (such as Alzheimer’s disease [35], Parkinson’s disease [36], Huntington’s disease [37], and amyotrophic lateral sclerosis [38]), cancer (such as lung cancer, gastric tumors, colon cancer, melanoma, ovarian cancer, Burkitt’s lymphoma [39,40], and others [41]), and cardiovascular diseases (e.g., myocardial infarction, cardiomyopathies and heart failure [42], Alport syndrome [43], Loeys–Dietz syndrome [44], and others). Changed levels of ncRNAs were found also in some LSDs, namely, in Niemmann–Pick disease type C (NPC) and Gaucher disease [2]. In NPC, where cholesterol is a stored material, miRNA can be involved in the regulation of the expression of genes coding for enzymes necessary for the metabolism of cholesterol and other lipids. Studies on fibroblasts derived from patients suffering from Gaucher disease indicated that some miRNAs can either enhance or inhibit the activity of glucocerebrosidase, an enzyme that is deficient in patients suffering from this disease. Thus, miRNAs can act as modulators of metabolism in these diseases [45,46,47,48]. This finding was the basis for the hypothesis that ncRNAs may be potential targets for novel therapeutics to be used against inherited diseases [49].

There is only very limited knowledge regarding the role of ncRNAs in MPS. The only studies conducted in this field and published recently concerned a mouse model of MPS I in which only one miRNA was investigated. It was demonstrated that the modulation of the level of miR-17 is responsible for the impaired expression of the *Neu1* gene [50]. This gene codes for neuraminidase 1, which is involved in the degradation of gangliosides [51], which are compounds that act as secondary storage material in MPS [6].

It is worth mentioning that studies on ncRNA in the pathogenesis of LSD—published to date—have focused solely on miRNA molecules. The role of lcnRNAs remains unknown, and it is definitely worth investigation as the influence of such RNAs on pathomechanisms has been demonstrated for various neurodegenerative diseases (see above). Thus, the aim of this work was to assess the potential role of lncRNAs in MPS. In this first (to our knowledge) study on lncRNAs in MPS, we employed a transcriptomic approach that should facilitate the demonstration of a global picture of the changes in the levels of these RNA species as well as the estimation of their potential roles in the pathomechanisms of these complex diseases. Moreover, it was assumed that studies on lncRNAs in MPS might help to explain the incomplete efficiency of therapies based solely on reducing GAG levels, and perhaps indicate new methods for the development of combined therapies that could involve the modulation of the levels of lncRNAs and/or miRNA, thus improving the control of the expression of specific genes, which is crucial for the effective abolition of all cellular defects occurring in MPS.

## 2. Materials and Methods

### 2.1. Lines of Human Fibroblasts and Cell Cultures

Patient-derived fibroblasts were used in this study. The cell lines were purchased from the NIGMS Human Genetic Cell Repository at the Coriell Institute for Medical Research (this Institute incorporates all required forms of bio-ethical approval). MPS cell lines were characterized by the presence of specific mutations or dysfunction of the corresponding enzyme (when mutations were not determined). Their features are presented in Table 1. HDFa cell line (healthy fibroblasts) was used as a control. Cells were cultured in the DMEM medium, which was supplemented with 10% fetal bovine serum. Standard mixture of antibiotics was added to this medium. Cultures of fibroblasts were incubated at 37 °C, with 5% CO_2_ saturation, and at 95% humidity.

### 2.2. Transcriptomic Analyses

Transcriptomic analyses were performed according to the previously described procedure (for statistical analyses, results from 4 independent biological experiments were used) [52]. Briefly, 5 × 10^5^ cells were withdrawn from cell cultures (passages between 4 and 15) and homogenized using the QIAshredder columns in the presence of guanidine isothiocyanate and β-mercaptoethanol. RNA was extracted with the RNeasy Mini kit (Qiagen, Hilden, Germany) and treated with Turbo DNase (Life Technologies, Life Technologies, Carlsbad, CA, USA). RNA quality was tested using RNA Nano Chips in the Agilent 2100 Bioanalyzer System (Agilent Technologies, Santa Clara, CA, USA). cDNA libraries were constructed based on mRNA libraries using the Illumina TruSeq Stranded mRNA Library Prep Kit (Illumina, San Diego, CA, USA). Sequencing was performed by employing HiSeq4000 (Illumina, San Diego, CA, USA). Raw reads (4 × 10^7^ from every single experiment) were analyzed, yielding over 12 Gb of raw data per sample. In order to map the raw readings, the GRCh38 human reference genome, taken from the Ensembl database (https://www.ensembl.org/, last accessed on 1 December 2022), was employed. For annotation and classification of transcripts, the BioMart interface was used. Raw data of the RNA-seq analysis have been deposited in the NCBI Sequence Read Archive (SRA) (accession no. PRJNA562649).

### 2.3. Statistical Analyses

In the transcriptomic analyses, statistical significance was calculated based on results obtained from 4 independent biological experiments (n = 4). For the primary assessment of values with normal continuous distribution, one-way ANOVA was used with log_2_(1 + x). To calculate the false discovery rate (FDR), the Benjamini–Hochberg method was used. To compare two groups, post hoc Student’s *t*-test was used with Bonferroni correction. These calculations were performed using the R software v3.4.3 (https://cran.r-project.org/bin/windows/base/old/3.4.3/, last accessed on 1 December 2022 last accessed on 5 December 2022). The differences were considered significant when *p* < 0.1, which is used as a standard in transcriptomic analyses [15,16,17,18,19,20,21,22,23].

## 3. Results

To investigate changes in the levels of lncRNAs in the MPS cells relative to the control fibroblasts, we conducted transcriptomic analyses using cell lines derived from patients suffering from most MPS types (I, II, IIIA, IIIB, IIIC, IIID, IVA, IVB, VI, VII, and IX). The RNA-seq studies were performed as described in Section 2. Although only one cell line per MPS type was used, every experiment was performed independently four times; thus, there were four biological repeats (n = 4). Since our goal was to investigate potential changes in the levels of lncRNAs in MPS patients for the first time, we also considered all MPS types as a group of diseases and assessed whether any changes were common for most or all types. This made our transcriptomic analyses reliable, as indicated in previously published articles [15,16,17,18,19,20,21,22,23]. For analyses concerning long RNA molecules, the standard RNA isolation procedure allowed for the purification of both mRNA and lncRNA species, which enabled us to compare the abundance of specific kinds of these nucleic acids in a single experiment.

Our global analysis of the abundance of RNA molecules indicated that there are changes in the levels of various lncRNAs in every MPS type. Between 2 (in MPS II) and 13 (in MPS IX) lncRNA transcripts were significantly changed (either up- or down-regulated) relative to the control fibroblasts (Figure 1). These results indicated that there can be potential changes in the lncRNA-mediated regulation of the expression of various genes in MPS cells. 

To assess which kinds of lncRNA are commonly up- or down-regulated in MPS cells, we assessed specific long non-coding transcripts that revealed either increased or decreased abundance relative to the control cells in fibroblasts of at least four MPS types. Five of such lncRNAs were identified, and are specified in Table 2. Importantly, in every case, a specific lncRNA was either up- or down-regulated in all MPS types wherein significant differences relative to the control cells occurred (no cases of up-regulation in some and down-regulation in other MPS types were identified). Therefore, it appears that similar changes in the levels of specific lncRNA occur in all/most of the MPS types.

In the next step, we analyzed how many and which lncRNAs occurred at especially increased or decreased levels in MPS cells. For this kind of analysis, we grouped lncRNAs into cohorts, revealing specific log_2_FC values (where FC states for fold change). These calculations indicated that the levels of most affected lncRNAs changed a few times; however, there were long non-coding transcripts that had been up- or down-regulated over ~6 times (log_2_FC > 2.5 or <−2.5) (Figure 2).

Specific genes encoding lncRNAs occurring at especially elevated or decreased levels (over ~6-fold, i.e., log_2_FC > 2.5 or <−2.5) in the MPS cells have been identified, and they are presented in Table 3. Again, the direction of the changes was always the same for each specific lncRNA type, i.e., every lncRNA was either up- or down-regulated in all MPS types. This confirms the uniform character of the changes in the levels of lncRNAs in MPS, irrespective of the disease type.

To identify potential target genes for specific lncRNA-mediated regulations, we employed the NPInter v4.0 data base (http://bigdata.ibp.ac.cn/npinter4/#, last accessed on 5 December 2022). Then, among all the genes potentially regulated by a given lncRNA, we selected those which revealed significant changes in at least one MPS type, as indicated by the results of our transcriptomic (RNA-seq) analyses for mRNAs (NCBI Sequence Read Archive (SRA) accession no. PRJNA562649). The results of these analyses are presented in Figure 3. They revealed that increased levels of the *GAS5* lncRNA correlate with the up-regulation of the *HNRNPC* gene in MPS IIIC, the down-regulation of the *FXR1* gene in MPS IIIC, and the down-regulation of the *MATR3* gene in MPS IIIB (Figure 3A). Higher abundance of the *MEG3* lncRNA occurred simultaneously with increased levels of *TP53* mRNA in MPS IIIC (Figure 3B) and decreased levels of *TARDBP* mRNA in MPS IIIB (Figure 3C). The down-regulation of *LINC00856* lncRNA correlated with the down-regulation of the *MATR3* gene in MPS IIIB (Figure 3D).

## 4. Discussion

Although MPS is a group of monogenic diseases characterized by the accumulation of GAGs in lysosomes, recent studies clearly indicated that primary storage is not the only cause of significant changes in cellular physiology. Among various secondary and tertiary defects that significantly contribute to the development of specific MPS symptoms in patients, the dysregulation of the expression of a battery of genes has been demonstrated to significantly influence the structures and functions of cellular organelles as well as different cellular processes [15,16,17,18,19,20,21,22,23]. Recent findings have suggested that changes in the expression of genes coding for regulators of activities of other genes might be responsible for triggering chains of processes that considerably contribute to dysfunctions of cells, tissues, organs, and entire organisms [16]. Since non-coding RNA molecules can modulate the ex-pression of hundreds or thousands of genes [24,25,26,27], we investigated whether the levels of these kinds of regulatory RNAs are changed in MPS cells. In this study, we have focused on lncRNAs, as this kind of non-coding RNAs was not investigated previously in MPS cells. 

In this report, RNA-seq analyses of the biological material of fibroblasts derived from patients suffering from most MPS types indicated, for the first time, that the levels of various lncRNAs are significantly changed relative to control cells (Figure 1, Table 2). Some lncRNAs were up- or down-regulated especially significantly (over 6-times; Figure 2, Table 3), and using the NPInter v4.0 data base, we identified potential target genes whose expression could be regulated by the actions of these non-coding transcripts. Then, by analyzing the levels of mRNAs derived from the same cells, genes that could be affected by specific lncRNAs and whose expression was up- or down-regulated were indicated. Therefore, we assume that the expression of these genes is likely modulated by changes in the levels of specific lncRNAs in MPS cells. The putative pairs of such lncRNAs and protein-encoding genes are as follows: *GAS5* lncRNA—*HNRNPC*, *FXR1*, and *MATR3*; *MEG3* lncRNA—*TP53* and *TARDBP*; and *LINC00856* lncRNA —*MATR3* (Figure 3). Interestingly, this putative lncRNA-mediated regulation was especially pronounced in different subtypes of Sanfilippo disease (MPS III).

One may ask: what are the roles of the genetic elements involved in the above-described regulations? It has been demonstrated that the *GAS5* lncRNA controls cancer development [53,54]. Its potential target genes, *HNRNPC* and *FXR1*, whose expression is changed in MPS cells, encode heterogeneous nuclear ribonucleoprotein C (a protein involved in the control of splicing and translation) [55] and RNA-binding protein fragile-X mental retardation autosomal 1 (a factor that participates in the mRNA transport from the nucleus and is bound to polysomes) [56,57], respectively. The *MATR3* gene product, matrin 3, is a protein that interacts with nucleic acids and is able to bind to both DNA and RNA fragments, though its exact functions are still poorly understood [58,59]. Like *GAS5*, the *MEG3* lncRNA is a tumor-growth inhibitor [60,61]. Potential *MEG3*-regulated genes, whose changed expression correlated with increased levels of this lncRNA in MPS cells, include *TP53* and *TARDBP*. These genes encode the major tumor-suppressor protein p53 [62] and TAR DNA-binding protein 43 (or TDP-43) (an RNA/DNA-binding protein that is involved in RNA transactions) [63], respectively. Previously, the *LINC00856* lncRNA was found to be involved in the response to acute lymphoblastic leukemia [64], and the down-regulation of *MATR3* correlated with decreased levels of this lncRNA in MPS cells.

Interestingly, as indicated in the preceding paragraph, genes strongly affected by lncRNAs in MPS cells are involved in the regulation of the expression of other genes, which is mainly performed through interactions with nucleic acids. It is, therefore, tempting to speculate that the dysregulation of the expression of regulatory genes might significantly contribute to the development of cellular dysfunctions in MPS cells. If such a dysregulation is strong enough, the cellular changes might become irreversible or hardly reversible. In such a case, the pathogenic cascade of cellular misfunctions would not be corrected by the simple removal of the primary storage material (GAGs). This might explain why various therapies for MPS fail to treat all the symptoms of patients, despite the normalization of GAG levels in body fluids [9,10,11]. In this light, specific lncRNAs might be considered as potential targets for novel auxiliary drugs against MPS. Namely, the primary therapeutic intervention (ERT, hematopoietic stem cell transplantation, gene therapy, or others) would cause the normalization of GAG levels, while the accompanying, lcnRNA-specific drug might modulate the expression of specific genes, leading to the corrections of otherwise hardly improved cellular processes. 

The observed correlations between the changes in the levels of lncRNAs in MPS cells and the expression efficiency of the target genes were either direct or reverse (Figure 3). Various lncRNAs can regulate gene expression by either sponging miRNAs (alleviating negative regulation) or interacting directly with mRNA molecules (modulating their abundance and translational efficiency) [33,34,65]. Therefore, when the directions of the changes in the levels of specific lncRNA and an mRNA of its target gene are the same, the lncRNA acts, most probably, through the interaction with miRNA(s) and the prevention of mRNA degradation. On the other hand, when the directions of changes are opposite, direct lncRNA-mRNA interactions are more likely.

This study, which demonstrates changes in levels of lncRNAs in MPS cells for the first time, also has some limitations. One of these was the use of only one cell line per MPS type. However, the biological experiments were repeated four times, which made the results of the RNA-seq analyses reliable [15,16,17,18,19,20,21,22,23]. The presented conclusions can also be corroborated by the fact that the directions of the changes in the levels of specific lncRNAs were always the same in all MPS types. Another limitation was that our study only focused on transcriptomic analyses. Indeed, further studies with different investigatory methods of lncRNA functions are necessary to understand the details of their roles in MPS. Nevertheless, this is the first report signaling the importance of changes in the levels of lncRNAs in MPS; thus, it can open a new field of research into understanding the molecular mechanisms of the pathogenesis of this disease.

## 5. Conclusions

For the first time, changed levels of various lncRNAs were demonstrated in 11 types/subtypes of MPS. Increased or decreased levels of specific lncRNAs correlated with the up- or down-regulation of the expression of genes that are potential targets for lncRNA-mediated control, suggesting functional relationships. These analyses suggest that changes in the abundance of lncRNAs can considerably influence the pathomechanism of MPS through the dysregulation of the expression of certain genes, especially those involved in the control of the activities of other genes.

## Figures and Tables

**Figure 1 genes-14-00271-f001:**
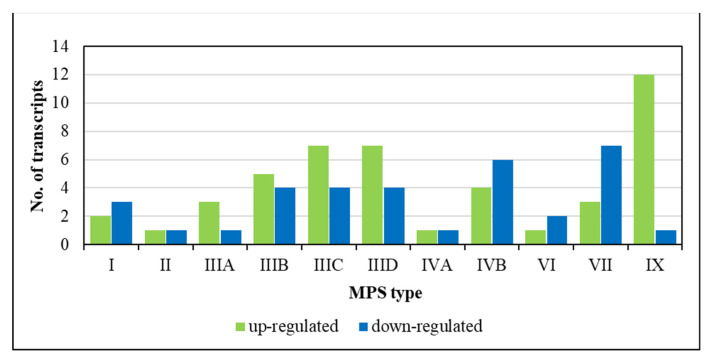
Number of up- and down-regulated transcripts of genes coding for lncRNAs (Gene Ontology term GO:0140742 (lncRNA transcription) in the QuickGO database (https://www.ebi.ac.uk/QuickGO/, last accessed on 5 December 2022)) in different types of MPS relative to control cells (HDFa).

**Figure 2 genes-14-00271-f002:**
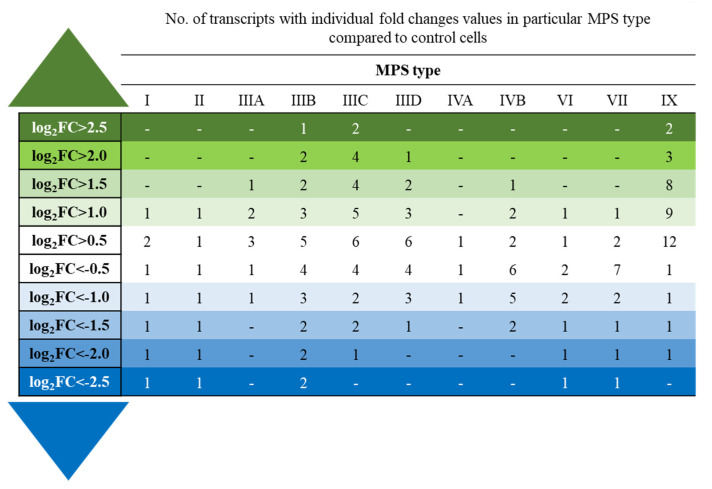
Number of transcripts included in the ‘lncRNA transcription’ term (GO:0140742) in the QuickGO database with altered expression depending on the level of fold-change (log2FC) in different types of MPS relative to control cells (HDFa).

**Figure 3 genes-14-00271-f003:**
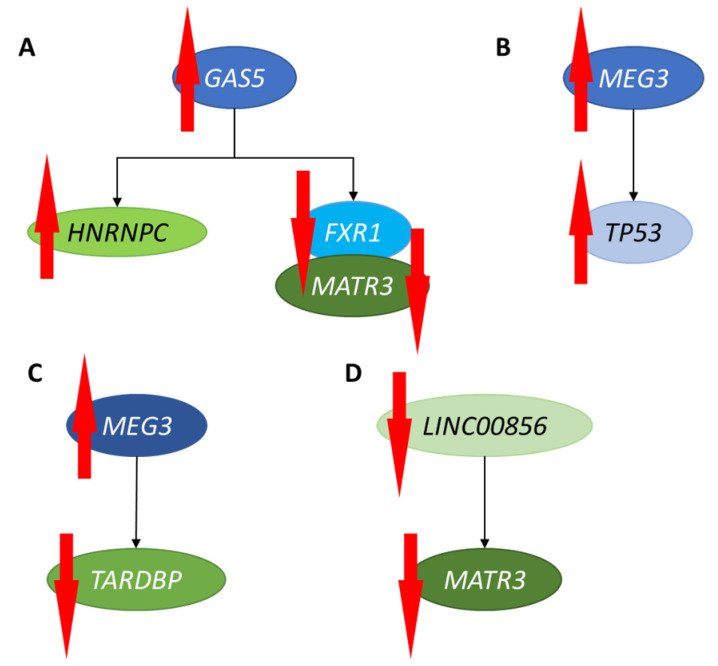
Correlations of increased (red, upward-pointing arrows) and decreased (red, downward-pointing arrows) levels of *GAS5* (**A**), *MEG3* (**B**,**C**), and *LINC00856* (**D**) lncRNAs with significant changes (either increased or decreased abundance of mRNAs, marked by upward- or downward-pointing red arrows, respectively) in expression of specific genes in MPS cells.

**Table 1 genes-14-00271-t001:** Fibroblast lines derived from MPS patients and used in this study.

MPS Type	Coriell Institute Catalogue Number	Sex of Patient	Age (Number of Years at the Time of Sample Collection)	Mutated Gene	Mutation(s)
MPS I	GM00798	F	1	*IDUA*	p.Trp402Ter/p.Trp402Ter
MPS II	GM13203	M	3	*IDS*	p.His70ProfsTer29/-
MPS IIIA	GM00879	F	3	*SGSH*	p.Glu447Lys/p.Arg245His
MPS IIIB	GM00156	M	7	*NAGLU*	p.Arg626Ter/p.Arg626Ter
MPS IIIC	GM05157	M	8	*HGSNAT*	p.Gly262Arg/p.Arg509Asp
MPS IIID	GM05093	M	7	*GNS*	p.Arg355Ter/p.Arg355Ter
MPS IVA	GM00593	F	7	*GALNS*	p.Arg386Cys/p.Phe285Ter
MPS IVB	GM03251	F	4	*GLB1*	p.Trp273Leu/p.Trp509Cys
MPS VI	GM03722	F	3	*ARSB*	Not determined
MPS VII	GM00121	M	3	*GUSB*	p.Trp627Cys/p.Arg356Ter
MPS IX	GM17494	F	14	*HYAL1*	Not determined

**Table 2 genes-14-00271-t002:** Genes encoding lncRNAs whose expression is significantly changed in at least four MPS types relative to the control cells. Changes in expression (transcripts’ levels) of particular genes in MPS types are depicted, indicating log_2_ fold change (FC) relative to control cells; up-regulation is marked in green while down-regulation in marked in blue; non-statistically significant differences are not colored.

Transcript	log_2_FC of Selected Transcripts’ Levels in at Least 4 MPS Types vs. HDFa Line										
	I	II	IIIA	IIIB	IIIC	IIID	IVA	IVB	VI	VII	IX
*GAS5*	0.83	1.16	1.26	0.93	2.98	0.11	0.36	0.69	1.35	−0.02	0.96
*ILF3-DT*	0.63	0.55	0.33	−0.34	0.79	0.68	0.77	0.31	0.81	0.01	0.82
*SNHG8*	0.76	−0.11	1.68	−0.62	1.06	1.25	0.33	−1.29	−0.46	−0.58	1.54
*LINC00667*	1.10	0.73	0.73	0.80	−0.08	−0.40	−0.49	1.10	−0.10	−1.14	1.23
*LINC02381*	−0.10	−0.57	−0.10	−1.39	−2.14	−1.58	−1.48	−1.39	−1.06	0.16	−1.27

**Table 3 genes-14-00271-t003:** Genes included in the ‘lncRNA transcription’ term (GO:0140742) with log_2_FC > 2.5 in specific MPS types relative to control cells (HDFa). Upward- and downward-pointing arrows indicate up- and down-regulation, respectively, while minus symbols indicate lower levels of changes. For *MEG3*, three transcripts were identified, which were marked as t.1, t.2, and t.3.

Transcript	log_2_FC > 2.5 or <−2.5 of Selected Transcripts’ Levels in Particular MPS Type vs. HDFa Line										
	I	II	IIIA	IIIB	IIIC	IIID	IVA	IVB	VI	VII	IX
*SNHG5*	↓	−	−	↓	−	−	−	−	−	↓	−
*LINC01705*	↓	↓	−	−	−	−	−	−	↓	−	−
*LINC00856*	−	−	−	↓	−	−	−	−	−	−	−
*CYTOR*	−	−	−	−	−	−	−	−	−	−	↓
*MEG3* (t.1)	−	−	−	↑	↑	−	−	−	−	−	−
*MEG3* (t.2)	−	−	−	−	−	−	−	−	−	−	↑
*MEG3* (t.3)	−	−	−	−	−	−	−	−	−	−	↑
*GAS5*	−	−	−	−	↑	−	−	−	−	−	−

## Data Availability

Raw data of the RNA-seq analysis are deposited in the NCBI Sequence Read Archive (SRA), under accession no. PRJNA562649.

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
