# Peer review of "Expression of Long Noncoding RNAs in Fibroblasts from Mucopolysaccharidosis Patients"

_genes, 2023, doi:10.3390/genes14020271_

Round 1

Reviewer 1 Report

In this manuscript, Ciske and collegues identified for the first time level changes in long non-coding RNAs (lncRNAs) using transcriptomic analyses in fibroblasts from mucopolysaccharidosis (MPS) patients. lncRNA SNHG5LINC01705LINC00856CYTORMEG3, and GAS5 were over 6-fold upregulated, compared to control cells, in several types of MPS. Some potential target genes for regulation by these lncRNAs were identified, and correlations between changed levels of specific lncRNAs and changes in abundance of mRNA transcripts of these genes (HNRNPC, FXR1TP53, TARDBP, and MATR3) were found. 

The results suggest that changes in lncRNAs levels can influence pathomechanism of MPS through dysregulation of expression of target genes involved in the control of activities of other genes.

The manuscript is of great interest in the field of Mucopolysaccharidosis. It is well written, organized and the results are clearly presented. 

Author Response

RESPONSE: We thank the reviewer for positive comments. We are glad that the reviewer evaluated our manuscript as being of a great interest. Since no critical comments were included, we did not introduce any changes in response to the report of this reviewer.

Reviewer 2 Report

In this paper the authors used RNA-seq analysis to identify several lncRNAs that are differentially expressed in fibroblasts from mucopolysaccharidosis (MPS) patients as compared to control fibroblasts. Next, potential targets genes regulated by these lncRNAs were identified by in silico analysis. Next, expression of the lncRNA and the target genes was corelated to glean insight into the pathomechanisms of MPS.

The major problem with this manuscript is the English language. The text contains numerous problems with punctuation, grammar, tense, typos, and/or sentence structure and clarity. Moreover, many sentences are written with an inappropriate colloquial/jargon style and often lack logical structure. These errors are too numerous to list, but some general suggestions are provided below.

1.     The title is very awkward and should be modified. Also, calling cell lines from MPS patients a “cellular model” is incorrect. A suggested title might be: “Expression of long noncoding RNAs in fibroblasts from mucopolysaccharidosis patients”.

2.     Introduction is too long and unfocused and should be streamlined.

3.     Materials and Methods do not describe the employed methodology in sufficient details. For example, what methods were used to construct cDNA libraries?

4.     The list of MPS fibroblasts is unwieldy and should be presented in a table.

5.     All items of equipment and software must be properly identified by manufacturers and their addresses. “R software v3.4.3” (line 163) is not enough.

6.     Results and Discussion sections are simply overlong, which has negative effects on clarity and general readability.

In summary, the manuscript needs substantial editorial and organizational changes that foreground the data critical to the authors' claims while removing extraneous or tangential information. Also, it would benefit from more precise and muted claims. The manuscript requires an extensive editing by a professional scientific English service provider before re-submission.

Author Response

REVIEWER’S COMMENT: In this paper the authors used RNA-seq analysis to identify several lncRNAs that are differentially expressed in fibroblasts from mucopolysaccharidosis (MPS) patients as compared to control fibroblasts. Next, potential targets genes regulated by these lncRNAs were identified by in silico analysis. Next, expression of the lncRNA and the target genes was corelated to glean insight into the pathomechanisms of MPS.

The major problem with this manuscript is the English language. The text contains numerous problems with punctuation, grammar, tense, typos, and/or sentence structure and clarity. Moreover, many sentences are written with an inappropriate colloquial/jargon style and often lack logical structure. These errors are too numerous to list, but some general suggestions are provided below.

RESPONSE: We are sorry for the English errors indicated by the reviewer. The manuscript has been modified according to the reviewer’s suggestions and checked by a person professional in English.

  1. REVIEWER’S COMMENT: The title is very awkward and should be modified. Also, calling cell lines from MPS patients a “cellular model” is incorrect. A suggested title might be: “Expression of long noncoding RNAs in fibroblasts from mucopolysaccharidosis patients”.

RESPONSE: We have followed the suggestion of the reviewer, and the modified title reads as recommended.

  1. REVIEWER’S COMMENT: Introduction is too long and unfocused and should be streamlined.

RESPONSE: Introduction has been shortened significantly (from 1242 to 816 words), and it is more focused now, according to the reviewer’s recommendation (see lines 24-90 in the revised manuscript). 

  1. REVIEWER’S COMMENT: Materials and Methods do not describe the employed methodology in sufficient details. For example, what methods were used to construct cDNA libraries?

RESPONSE: According to the reviewer’s recommendation, the materials and methods are described in more details, as indicated in section 2.2 (lines 103-119).

  1. REVIEWER’S COMMENT: The list of MPS fibroblasts is unwieldy and should be presented in a table.

RESPONSE: As requested by the reviewer, lines of MPS fibroblasts are presented in a table (Table 1 in the revised manuscript).

  1. REVIEWER’S COMMENT: All items of equipment and software must be properly identified by manufacturers and their addresses. “R software v3.4.3” (line 163) is not enough.

      RESPONSE: As requested by the reviewer, manufacturers with addresses or web-page addresses are provided for each instrument, kit, and software.

  1. REVIEWER’S COMMENT: Results and Discussion sections are simply overlong, which has negative effects on clarity and general readability.

RESPONSE: As recommended by the reviewer, Results and Discussion sections were shortened considerably (from about 4 pages to 3 pages, and from almost 2 pages to slightly over 1 page, respectively), and are more focused now.

REVIEWER’S COMMENT: In summary, the manuscript needs substantial editorial and organizational changes that foreground the data critical to the authors' claims while removing extraneous or tangential information. Also, it would benefit from more precise and muted claims. The manuscript requires an extensive editing by a professional scientific English service provider before re-submission.

RESPONSE: We have followed all reviewer’s recommendations. Editorial and organizational changes have been introduced to make the paper more focused. Conclusions are more precise and toned down now. The text has been professionally checked for English usage.